# Seasonal variation of nitryl chloride and its relation to gas-phase precursors during the JULIAC campaign in Germany

Zhaofeng Tan[1], Hendrik Fuchs[1,4], Andreas Hofzumahaus[1], William J. Bloss[3], Birger Bohn[1], Changmin Cho[1], Thorsten Hohaus[1], Frank Holland[1], Chandrakiran Lakshmisha[1], Lu Liu[1], Paul S. Monks[2], Anna Novelli[1], Doreen Niether[1], Franz Rohrer[1], Ralf Tillmann[1], Thalassa Valkenburg[2], Vaishali Vardhan[1,a], Astrid Kiendler-Scharr[1,4], Andreas Wahner[1], Roberto Sommariva[2,3]

[1] Institute of Energy and Climate Research, IEK-8: Troposphere, Forschungszentrum Jülich GmbH, Jülich, Germany

[2] School of Chemistry, University of Leicester, Leicester, UK

[3] School of Geography, Earth and Environmental Sciences, University of Birmingham, Birmingham, UK

[4] Physikalisches Institut, Universität zu Köln, Köln, Germany

[a] now at, Department of Chemistry, University College Cork, Ireland

Correspondence: Zhaofeng Tan (zh.tan@fz-juelich.de) and Roberto Sommariva (rs445@le.ac.uk)

**Abstract.** Ambient measurements of nitryl chloride ($ClNO_2$) were performed at a rural site in Germany covering 3 periods in winter, summer, and autumn 2019 as part of the JULIAC campaign (Jülich Atmospheric Chemistry Project) that aimed for understanding the photochemical processes in air masses typical for mid-west Europe. Measurements were conducted at 50 m above ground, which was mainly located in the nocturnal boundary layer and thus uncoupled from local surface emissions. $ClNO_2$ is produced at nighttime by heterogeneous reaction of dinitrogen pentoxide ($N_2O_5$) on chloride ($Cl^-$) containing aerosol. Its photolysis during the day is of general interest as it produces chlorine (Cl) atoms that react with different atmospheric trace gases forming radicals. The highest observed $ClNO_2$ mixing ratio was 1.6 ppbv (15-min average) during the night of September 20. Air masses reaching the measurement site either originated from long-range transport from the southwest and had an oceanic influence or circulated in the nearby region and were influenced by anthropogenic activities. Nocturnal maximum $ClNO_2$ mixing ratios were around 0.2 ppbv if originating from long-range transport in nearly all seasons, while values were higher ranging from 0.4 to 0.6 ppbv for regionally influenced air. The chemical composition of long-range transported air was similar in all investigated seasons, while the

regional air exhibited larger differences between the seasons. The $N_2O_5$ necessary for $ClNO_2$ formation
comes from the reaction of nitrate radicals ($NO_3$) with nitrogen dioxide ($NO_2$), where $NO_3$ itself is formed
by reaction of $NO_2$ with ozone ($O_3$). Measured concentrations of $ClNO_2$, $NO_2$ and $O_3$ were used to
quantify $ClNO_2$ production efficiencies, i.e., the yield of $ClNO_2$ formation per $NO_3$ radical formed, and a
box model was used to examine the idealized dependence of $ClNO_2$ on the observed nocturnal $O_3$ and
$NO_2$ concentrations. Results indicate that $ClNO_2$ production efficiency was most sensitive to the
availability of $NO_2$ rather than that of $O_3$ and increase with decreasing temperature. The average $ClNO_2$
production efficiency was highest in February and September with values of 18% and was lowest in
December with values of 3%. The average $ClNO_2$ production efficiencies were in the range of 3 and 6 %
from August to November for air masses originating from long-range transportation. These numbers are
at the high end of values reported in literature indicating the importance of $ClNO_2$ chemistry in rural
environments in mid-west Europe.

## 1 Introduction


Nitryl chloride ($ClNO_2$) is an important nocturnal reservoir for nitrogen oxides (Brown and Stutz, 2012),
because it accumulates during the night and photolyzes to nitrogen dioxide ($NO_2$) and a chlorine atom
(Cl) after sunrise in the morning (Reaction R1).
$ClNO_2 + hv \rightarrow NO_2 + Cl$ (R1)
Chlorine atoms are a highly reactive oxidant in the atmosphere, initiating, for example, the degradation of
volatile organic compounds (VOCs) and thereby contributing to the formation of ozone ($O_3$) and other
pollutants (Simpson et al., 2015;Thornton et al., 2010;Mielke et al., 2011;Young et al., 2012). In some
studies, $ClNO_2$ was shown to increase the daily ozone production from sub ppbv levels to mixing ratios of
up to 10 ppbv, so that $ClNO_2$ chemistry contributed substantially to photochemical ozone production
(Osthoff et al., 2008;Wang et al., 2016;Sommariva et al., 2021).
$ClNO_2$ formation is initiated by the heterogeneous reaction of dinitrogen pentoxide ($N_2O_5$) on aqueous
surfaces that contains chloride ($Cl^-$) (Roberts et al., 2009;George and Abbatt, 2010;Osthoff et al.,
2008;Thornton et al., 2010). The entire chemical reaction chain is described as McDuffie et al. (2018a):
$NO_2 + O_3 \rightarrow NO_3 + O_2$ (R2)
$NO_3 + NO_2 \rightarrow N_2O_5$ (R3a)
$N_2O_5 \rightarrow NO_3 + NO_2$ (R3b)
$N_2O_{5\,(g)} + aerosol_{(aq,Cl^-)} \rightarrow \varphi \times \left(ClNO_{2\,(g)} + HNO_{3\,(aq)}\right) + (1 - \varphi) \times 2\,HNO_{3\,(aq)}$          (R4)
where $\varphi$ is the yield ( $0 \le \varphi \le 1$) of gaseous $ClNO_2$ when $N_2O_5$ is taken up by aerosol.
$NO_3 + VOCs \rightarrow$ prod.                                                       (R5)
At night, nitrate radicals ($NO_3$) are produced by reaction of $NO_2$ with $O_3$ (Reaction R2), which then react
with another $NO_2$ to form $N_2O_5$ (Reaction R3a). $N_2O_5$ decomposes thermally back to $NO_2$ and $NO_3$
(Reaction R3b). The forward and back reactions constitute a fast thermal equilibrium between $NO_3$ and
$N_2O_5$ that is established  quickly established at temperatures typically found in the lower troposphere
(Brown and Stutz, 2012). Uptake of $N_2O_5$ on aqueous aerosol produces $ClNO_2$, when the particulate
phase of the aerosol contains dissolved chloride. The yield ($\varphi$) of $ClNO_2$ is a complex function of various
parameters such as temperature, aerosol water content, and chemical composition of the aerosol that
influence both uptake of $N_2O_5$ into the particles (McDuffie et al., 2018a) and the subsequent aqueous
phase chemistry leading to the formation of $ClNO_2$ (McDuffie et al., 2018a). The uptake of $N_2O_5$
(Reaction R4) and the reaction of $NO_3$ with VOCs (Reaction R5) constitute an overall loss term for the
sum of $NO_3$ and $N_2O_5$, because the fast equilibrium between $NO_3$ and $N_2O_5$. $HNO_3$ formation by Reaction
R4 is an important atmospheric sink for atmospheric nitrogen oxides in the lower atmosphere, because
$HNO_3$ photolysis is slow so that most of the produced $HNO_3$ does not reform $NO_2$, but is removed from
the atmosphere by deposition (Brown and Stutz, 2012). During the daytime, $NO_3$ is destroyed by
photolysis or by reaction with nitric oxide (NO). The thermal equilibrium between $NO_3$ and $N_2O_5$ thus
leads to a rapid depletion of $N_2O_5$ at day. Therefore, significant concentrations of $N_2O_5$ (the precursor of
$ClNO_2$) are usually only present at night.
Previous studies reporting $ClNO_2$ measurements in North America (Osthoff et al., 2008;Thornton et al.,
2010;Mielke et al., 2011;Wagner et al., 2012;Young et al., 2012;Mielke et al., 2013;Riedel et al.,
2013;McDuffie et al., 2018b;McNamara et al., 2020), Asia (Tham et al., 2016;Wang et al., 2016;Liu et
al., 2017;Wang et al., 2017a;Wang et al., 2017b;Le Breton et al., 2018;Yun et al., 2018;Zhou et al.,
2018;Yan et al., 2019;Jeong et al., 2019;Lou et al., 2022) and Europe (Phillips et al., 2012;Bannan et al.,
2015;Priestley et al., 2018;Sommariva et al., 2018) have shown that $ClNO_2$ is present in various
environments even at a distance from the coast, indicating that other sources of chloride than sea spray
contribute to the availability of chlorine for the formation of $ClNO_2$. Observed mixing ratios of $ClNO_2$ in
the atmosphere range from a few hundred pptv to several ppbv exhibiting significant spatial and temporal
variations.
Despite the large variation in $ClNO_2$ concentrations and its potentially important contribution to
photochemistry, systematic investigations of seasonal differences of $ClNO_2$ concentrations are sparse,
because $ClNO_2$ is not regularly measured at monitoring stations, but during intensive field campaigns,
which last typically only a few weeks. Sommariva et al. (2018) reported $ClNO_2$ measurements at three
different locations in the United Kingdom in all four seasons showing a clear seasonal variation with
maximum concentrations in spring and winter. Another study by Mielke et al. (2016) reporting the
seasonal behavior of $ClNO_2$ in Calgary, Canada, also showed maximum mixing ratios of $ClNO_2$ of up to
330 pptv in winter and spring.
The large variability of $ClNO_2$ concentrations in the atmosphere is due to the complexity of its formation
mechanism (Reactions R2 – R5) and the variability of its precursor concentrations. Assuming steady state
for the sum of $NO_3$ and $N_2O_5$ concentrations, the following relationship holds
$$\frac{d[NO_3+N_2O_5]}{dt} \cong 0 = k_2[NO_2][O_3] - k_{NO_3}[NO_3] - k_4[N_2O_5] \tag{Eq. 1}$$
where $k_{NO_3}$ represents the pseudo first-order rate constant for $NO_3$ loss mainly dominated by reactions
with atmospheric VOCs (Reaction R5) at night with no fresh NO emissions. Considering the thermal
equilibrium between $NO_3$ and $N_2O_5$, the $[NO_3]$ can be replaced by $[N_2O_5]/(K_{eq}(T)[NO_2])$ where $K_{eq}(T)$
is temperature dependent and equals to the ratio of the reaction rate constants of the thermal equilibrium,
i.e. $k_{3a}$ to $k_{3b}$ (Reaction R3a and b). Equation 1 can be solved for
$$[N_2O_5] = \frac{K_{eq}(T)[NO_2]}{k_{NO3} + K_{eq}(T)[NO_2]k_4} \cdot k_2[NO_2][O_3] \tag{Eq. 2}$$
The production rate of $ClNO_2$ is then
$$P_{ClNO2} = \varphi \cdot k_4 \cdot [N_2O_5] = \varphi \cdot \left(\frac{K_{eq}(T)[NO_2]k_4}{k_{NO_3} + K_{eq}(T)[NO_2]k_4}\right) \cdot k_2[NO_2][O_3] \tag{Eq. 3}$$
A production efficiency ε for $ClNO_2$ can be defined from this relationship
$$\epsilon_{ClNO2} = \frac{P_{ClNO2}}{k_2[NO_2][O_3]} = \varphi \left(\frac{K_{eq}(T)[NO_2]k_4}{k_{NO3} + K_{eq}(T)[NO_2]k_4}\right) \tag{Eq. 4}$$
It represents the formation rate of $ClNO_2$ from aerosol per $NO_3$ produced by the reaction of $NO_2$ with $O_3$
in the gas phase. Equations Eq. 3 and Eq. 4 describe the expected influences on the $ClNO_2$ formation by
its precursors $NO_2$ and $O_3$, by temperature and $NO_2$ controlling the equilibrium between $NO_3$ and $N_2O_5$,
and the competing loss reactions of $NO_3$ and $N_2O_5$ via Reactions R5 and R4, respectively. $\varphi$ is an
additional variable depending on the properties of the aerosol and specifically on its chloride content, as
mentioned above.
This study presents $ClNO_2$ measurements performed during the Jülich Atmospheric Chemistry Project
(JULIAC) campaign in three seasons (i.e. winter, summer, and autumn 2019). The JULIAC campaign
aimed to investigate the seasonal and diurnal variations of the atmospheric oxidation capacity at a rural
site that is typical for mid-west Europe. To minimize the impact of emissions from local sources, the air
was drawn from 50 m above ground ensuring that air is sampled from above the surface layer during
nighttime and flowed through the large environmental chamber SAPHIR at Forschungszentrum Jülich,
Germany. In this work, the seasonal variation of $ClNO_2$ concentrations and its formation are investigated.
As mentioned above, previous studies have demonstrated that $ClNO_2$ concentrations show significant
seasonal variations (Mielke et al., 2016; Sommariva et al., 2018). However, intensive seasonal
measurements in central Europe, to our knowledge, have not been performed so far. Given the ubiquitous
nature of $ClNO_2$ and its importance to enhance atmospheric oxidation processes, more detailed studies are
needed to broaden our knowledge of atmospheric $ClNO_2$ levels, its seasonal behavior and its distribution
in environments with different chemical conditions. In addition, this work presents empirical production
efficiencies of $ClNO_2$ determined from the nighttime measurements of $ClNO_2$, $NO_2$ and $O_3$ and analyzed
for their seasonal variations and origin of air masses, a prerequisite to understand the contribution of
$ClNO_2$ to radical photochemistry under the chemical and meteorological conditions encountered in this
campaign. Finally, a chemical box model is used here to understand the dependence of $ClNO_2$ formation
and production efficiency on the observed nocturnal $O_3$ and $NO_2$ concentrations. The measurements and
analysis presented in this paper help to illustrate the seasonal variability of $ClNO_2$ concentrations and
shed light on the factors that control its production in different seasons.

## 2    Methods

### 2.1 The JULIAC campaign

The JULIAC campaign was conducted in 2019 in the atmospheric simulation chamber SAPHIR on the
campus of Forschungszentrum Jülich, which is located at a rural site in Germany (50.91° N, 6.40° E).
The SAPHIR chamber consists of a double-wall Teflon film (volume: $(277 \pm 3)$ m$^3$) (Bohn et al.,
2005;Rohrer et al., 2005). Its high volume to surface ratio (1 m$^2$/m$^3$) minimizes air-surface interactions
within the chamber. The time scale of mixing is about 1 minute ensured by two fans that are operated
inside the chamber.
During this study, ambient air was drawn from 50 m height above ground into the chamber (Fig. S1,
Supporting Information). At this height, the air is expected to be decoupled from the surface layer during
the night, so that the air composition is not directly impacted by sources at the ground or deposition of
trace gases to the Earth's surface (Section 3.3). The inlet line (SilcoNert® coated stainless steel, inner
diameter: 104 mm) was mounted at a tower (JULIAC tower) next to the chamber. A fast flow rate of 660
$m^3$/h resulted in a residence time of the air inside the inlet line of approximately 4 s. The short residence
time and the inertness of the Silconert coating of the inlet line minimized loss and chemical changes of
the air before entering the SAPHIR chamber. The potential loss of trace gases in the inlet line was tested
for $O_3$, NO, $NO_2$, and CO and was found to be less than 5%.
Instruments could either sample air directly from the inlet line or the chamber volume. In the latter case,
part of the total air drawn through the inlet at the JULIAC tower flowed through the SAPHIR chamber
with a flow rate of 250 $m^3$/h that was controlled by a three-way valve right upstream of the injection point
into the chamber. The remaining part was vented. The residence time of sampled ambient air inside the
SAPHIR chamber was 1.1 hours calculated from the measured flow rate and the chamber volume.
Sampling air from the large volume of the SAPHIR chamber has the advantage that short-term variations
of trace gas concentrations flowed into the chamber for example due to local emissions or fast changes of
air masses are smoothed.
The JULIAC campaign consisted of four intensive measurement periods in winter (14 January to 10
February 2019), spring (08 April to 05 May 2019), summer (05 August to 01 September 2019), and
autumn (28 October to 24 November 2019). During these parts of the campaign, a large set of instruments
sampled air from the chamber. In addition, between each intensive measurement period, a limited set of
instruments for the detection of $ClNO_2$, $O_3$, NO, $NO_2$, OH reactivity, and VOCs continued measuring
directly from the inlet line at the JULIAC tower (Fig. S1, Supporting Information).

## 2.2 Instrumentation

A large set of instruments was deployed during the JULIAC campaign. In this work, the focus is on
measurements that are relevant to study the chemistry of $ClNO_2$.
$ClNO_2$ was measured by a chemical ionization mass spectrometry (CIMS) instrument from Leicester
University (THS Instruments LLC, GA, USA) operated in negative ion mode using iodide ($I^-$) as reagent
ion. $ClNO_2$ was detected at the mass to charge ratios (m/z) of 208 and 210 amu, corresponding to the two
isotopes of the $[I·ClNO_2]^-$ ion clusters as described in Sommariva et al. (2018).
The CIMS instrument was calibrated by standard additions of $ClNO_2$ generated by flowing humidified air
containing $Cl_2$ (from a cylinder containing a mixture of 5 ppmv (±5%) $Cl_2$ in $N_2$, Linde AG) over a salt
bath containing a 1:1 mixture of NaCl and $NaNO_2$ (Sommariva et al., 2018). The resulting $ClNO_2$
concentration in the air was determined by measuring the $NO_2$ concentration after thermally decomposing
$ClNO_2$ to Cl and $NO_2$ in a glass tube heated to a temperature of 400 °C. The $NO_2$ concentrations were
measured using a commercial $NO_2$ analyzer that makes use of the cavity attenuated phase-shift method
(CAPS, T500U, Teledyne API). The accuracy of $NO_2$ measurements by this analyzer is ±5%. The overall
accuracy of the $ClNO_2$ calibration is ±17%; the precision of $ClNO_2$ measurements is 13% with a 2-σ
detection limit of 5.6 pptv at a 1-minute time resolution.
The CIMS detection sensitivity depends on humidity because iodide ions form clusters with water
$(I·(H_2O)^-)$. The water-iodine cluster is a more efficient reagent ion for producing $I·(ClNO_2)^-$ clusters than
the $I^-$ ion (Kercher et al., 2009). The dependence of the sensitivity on humidity was characterized with
calibration experiments by varying the mixing ratios of water vapor. These experiments show that the
sensitivity of the instrument for the detection of $ClNO_2$ decreases by 19% per 1% water vapor mixing
ratio (Fig. S2, Supporting Information), when the signal is normalized to the $I·(H_2O)^-$ cluster signal (m/z
= 145). Calibrations of the instrument were performed during each measurement period using comparable
average humidity to that of the ambient air. The variability of the sensitivity due to the changes in
humidity in each 4-week long measurement period was less than ±5%. This is within the range of
reproducibility of calibration measurements. Therefore, the sensitivity was not corrected for the humidity
effect for individual data points, but an average sensitivity value was applied to all data from the entire
measurement period. The uncertainty due to the humidity dependence of the sensitivity and the
reproducibility of the calibration adds to the overall accuracy of $ClNO_2$ measurements increasing the
value to ±27%.
Photolysis frequencies inside the SAPHIR chamber were calculated from the actinic flux measured
outside the chamber and corrected for the reduction of radiation by shading effects and the transmission
of the Teflon film (Bohn et al., 2005). Ozone was detected by a UV photometer (model O342M, Ansyco).
Nitric oxide (NO) was measured by a chemiluminescence instrument (780TR, Eco Physics) that was also
used to detect $NO_2$ by conversion of $NO_2$ to NO in a blue-light photolytic converter upstream of the NO
analyzer. For the period after 01 December 2019, $NO_2$ was measured by an instrument using the iterative
cavity enhanced differential optical absorption spectroscopy method (ICAD1005, AirYX). The $NO_2$
measurements from the two instruments agreed well within 5%, when both instruments measured
concurrently. Water vapor and carbon monoxide (CO) concentrations were measured by a cavity ring-
down instrument (G2401, Picarro). $NO_3$ and $N_2O_5$ were measured by a custom-built cavity ring-down
instrument that is similar to the one described in Wagner et al. (2011).
Particle number concentration (for particles with a diameter > 5 nm) and size distribution (for particles
with a diameter between 10 and 1000 nm) were measured by a condensation particle counter (model
3787, TSI) and a scanning mobility particle sizer (model 3080, TSI), respectively. The aerosol surface
area ($S_a$) was calculated based on the particle number and geometric diameter in each size bin. The
chemical composition of particles was analyzed by an aerosol mass spectrometer (HR-TOF-AMS,
Aerodyne).
Temperature and pressure of the ambient air were measured inside the chamber and also outside the
chamber at different heights (2 m, 20 m, 30 m, 50 m, 80 m, 120 m) by sensors mounted at a
meteorological tower located approximately 200 m away from the SAPHIR chamber.

## 217    2.3 Comparability of measurements from the chamber and the inlet line

Air was sampled from 50 m above the ground from the top of the JULIAC tower at all times of the
campaign (Fig. S1, Supporting Information). However, $ClNO_2$ concentrations were determined in the air
from either one of the two sampling points during the different periods of the campaign.
During the intensive measurement periods (i.e. in February, August, and November), air was directly
sampled from the SAPHIR chamber. During other times, air was sampled from the inlet system of the
chamber at the JULIAC tower. In both cases, measured concentrations are representative for the air from
50 m height. In the case of sampling from the chamber, concentrations are averaged due to the 1-hour
residence time of air in the chamber.
To make data derived from both sampling points comparable, $ClNO_2$ concentrations measured inside the
chamber ($C_{chamber}$) were converted to equivalent concentrations at the tip of the JULIAC inlet system
($C_{50m}$). This can be achieved from the differential equation of concentrations taking into account dilution
with the flow rate ($k_{flow}$) and loss ($L_{chamber}$) and production ($P_{chamber}$) inside the chamber:
$\frac{dC_{chamber}}{dt} = k_{flow}(C_{50m} - C_{chamber}) + P_{chamber} - L_{chamber}$          (Eq. 5)
The concentration in the incoming air can be iteratively determined from the time series of measured
concentrations inside the chamber, if loss and production processes can be quantified. The other species
used in this work (e.g. O3, NOx, etc.) were measured both at the tip of the JULIAC inlet and inside
SAPHIR. Unless otherwise specified, the measurements presented in this work were taken at the tip of the
JULIAC inlet, or corrected using Eq 5.
Production of $ClNO_2$ from the heterogeneous reaction of $N_2O_5$ on particles is expected to be negligible on
the time scale of the residence time of air in the chamber for conditions of the JULIAC campaign.
Chamber wall interaction could be relevant because the surface area of the Teflon film is $10^6$ $\mu m^2/cm^3$,
i.e. several orders of magnitude larger than the surface area of ambient aerosol experienced in this
campaign, which were on the order of tens to hundreds of $\mu m^2/cm^3$. To quantify potential chamber
related loss and production processes, chamber characterization experiments were conducted (Section
3.1). They were analyzed by using a chemical box model, in which loss and production rates were
adjusted to reproduce measured $ClNO_2$ concentrations during these experiments. Temperature, relative
humidity, pressure, photolysis frequencies, and dilution rates determined from the air replenishment flow
rate were constrained to measurements in the model. The conversion of $N_2O_5$ to $ClNO_2$ via surface
reactions (Reaction R6) and the loss reactions of $ClNO_2$ on the chamber wall (Reaction R7) were included
in the model assuming pseudo-first order processes:
$N_2O_5 + wall \rightarrow ClNO_2$ $\hspace{6cm}$ (R6)
$ClNO_2 + wall \rightarrow products$ $\hspace{5.5cm}$ (R7)
In addition, the chemical loss of $ClNO_2$ via photolysis (Reaction R1) was considered. The results of these
experiments and the model analysis are discussed in Section 3.1.

## 3    Results and Discussion

### 3.1 Chamber effects on measured $ClNO_2$ concentrations

Two types of experiments were performed to characterize the chamber properties with respect to wall
interaction of $NO_3$, $N_2O_5$, and $ClNO_2$. In these chamber characterization experiments, only a small
replenishment flow of pure synthetic air compensated for leakages and extraction of air by instruments.
This led to a low dilution of trace gases with a rate that is equivalent to a lifetime of 17 hours in contrast
to the 1-hour lifetime during the operation of the chamber in the JULIAC campaign.
Three experiments were conducted (05, 06, and 07 February 2019) to test whether $ClNO_2$ was exclusively
lost by photolysis in the chamber or whether other processes, such as wall loss, contributed to the $ClNO_2$
removal. These experiments started with flowing ambient air through the SAPHIR chamber during the
night like in the operational mode of the JULIAC campaign (Section 2.1). The high flow was stopped
before sunrise (around 06:00 UTC) and the small replenishment flow was started. The evolution of trace
gas concentrations was observed until around 12:00 UTC while the air was exposed to sunlight. The $N_2O_5$
concentration decreased rapidly to zero after sunrise and thus no further $ClNO_2$ could be produced from
$N_2O_5$ conversion and $ClNO_2$ concentrations also decayed during the morning.
Measured concentrations are compared to calculation using a chemical box model (Section 2.3)
considering losses of $ClNO_2$ by dilution, photolysis, and potential wall loss. Whereas loss rates for
dilution and photolysis are constrained to measurements, the wall loss rate constant is adjusted to match
observed $ClNO_2$ concentrations. This results in a wall loss rate constant for $ClNO_2$ of $2.1 \times 10^{-5}$ $s^{-1}$. This
value is on the same order of magnitude as the loss rate constant of $ClNO_2$ due to photolysis ($4.1 \times 10^{-5}$ $s^{-1}$
at noon) and dilution ($1.5 \times 10^{-5}$ $s^{-1}$) for the experimental conditions of the characterization experiments.
Due to the higher chamber flow rate used during the JULIAC campaign, the dilution rate is an order of
magnitude higher ($2.5 \times 10^{-4}$ $s^{-1}$) than during the characterization experiments. Therefore, the wall loss rate
is only 8% of the dilution rate, and thus can be neglected in the further data analysis.
Additional three experiments were performed to characterize potential $ClNO_2$ formation from
heterogeneous reactions of $N_2O_5$ on the chamber wall. In these experiments (18 September, 18 October,
and 19 November 2019), $NO_2$ and $O_3$ were added into the dark chamber filled with pure, dry, or
humidified synthetic air. These experiments lasted for about 10 hours to observe the decay of $NO_2$ and $O_3$
concentrations and the accumulation of $ClNO_2$.
Fig. 1 shows measured concentrations for the experiments performed on 19 November. In this
experiment, the chamber air was humidified (RH=60%) and 28 ppbv of $NO_2$ and 80 ppbv of $O_3$ were
injected to produce $NO_3$ and $N_2O_5$. $NO_3$ mixing ratios were below the limit of detection (about a few
pptv) of the cavity ring-down instrument.
$N_2O_5$ measurements reached maximum mixing ratios of 0.17 ppbv shortly after the $O_3$ injection and
decreased afterward (Fig. 1). Also, $ClNO_2$ production was observed shortly after the ozone addition, when
$N_2O_5$ was present. Because the air was particle-free, one possible explanation for the formation of $ClNO_2$
is heterogeneous reaction of $N_2O_5$ on the chamber wall that may contain chloride, which could have been
deposited, for example, during previous experiments with ambient air.

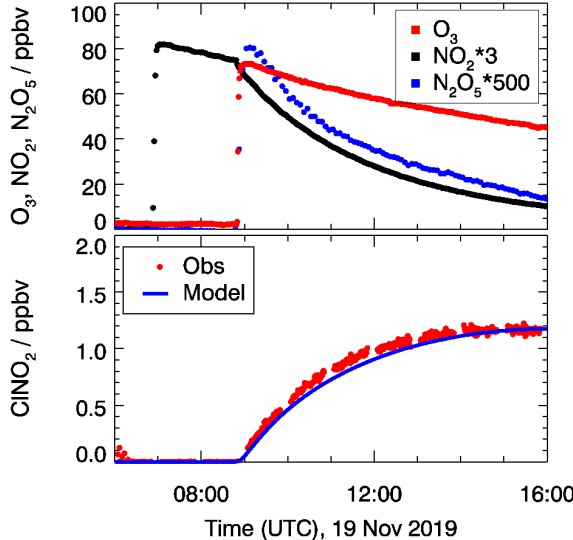


**Figure 1.** Chamber experiment to characterize $ClNO_2$ production from $N_2O_5$ conversion on the chamber wall in the dark on 19 November 2019. $ClNO_2$ concentrations are compared to model calculations taking conversion from $N_2O_5$ to $ClNO_2$ (Reaction R6) into account. A reaction rate constant of $8.2 \times 10^{-6}$ s$^{-1}$ is required to reproduce measured $ClNO_2$ concentrations.

The values of the conversion rates from $N_2O_5$ to $ClNO_2$ (Reaction R6) that are required to match the measured $ClNO_2$ concentrations in the model calculations are $k_{R6} = 4.0 \times 10^{-6}$ s$^{-1}$, $2.0 \times 10^{-6}$ s$^{-1}$, and $8.2 \times 10^{-6}$ s$^{-1}$ for the experiments on 18 September, 18 October and 19 November 2019, respectively.

During the JULIAC campaign, however, the potential contribution of $ClNO_2$ formation from $N_2O_5$ conversion on the chamber film was negligible. Taking the typical nocturnal $N_2O_5$ mixing ratio of about 50 pptv, the expected $ClNO_2$ production rate from $N_2O_5$ conversion on the chamber wall was about 1.5 pptv/h using the upper limit value of $k_6$ derived from the characterization experiments. This is less than 1% of the ambient $ClNO_2$ mixing ratio of up to several hundred pptv in the ambient air that is flowed into the chamber. Therefore, no corrections are needed for the interpretation of $ClNO_2$ measurements in the chamber.

Overall, the results of the characterization experiments allow to simplify the back-calculation of the $ClNO_2$ concentrations in the sampled air from measured concentrations in the chamber (Eq. 5). The chemical production rates and the deposition rates for $ClNO_2$ and $N_2O_5$ on the chamber walls can be neglected and only photolysis needs to be considered as a destruction process for $ClNO_2$ during the daytime. For nighttime conditions, $ClNO_2$ concentrations in the incoming air can be determined solely from the flow rate and the measured $ClNO_2$ concentration inside the chamber.

## 3.2 Overview of measurements

In order to determine the origin of air masses sampled at the measurement site, back trajectories were calculated using the HYSPLIT model (Stein et al., 2015) for every second hour. They were calculated for a height of 50 m above the ground and started 48 hours earlier before the air arrived at the measurement site. Calculations for different heights (500 m and 1000 m) gave similar results as the trajectories calculated for a height of 50 m. To extract information about the relation between the source of air masses and the measurements, the cluster analysis tool of the HYSPLIT model was used, which classified the trajectories into two groups (Fig. 2).

Trajectories showed most often prevailing long-distance transport of air masses from the southwest, from which they traveled hundreds of kilometers from the Atlantic Ocean (approximately 1000 km away from the measurement site) within 48 hours. These air masses were likely influenced by marine and continental emissions as they crossed over northern France and Belgium. They are referred to hereafter as belonging to the long-range transport group. The other group of trajectories did not show a prevalent direction but shared the common feature that these air masses circulated over the cities nearby the measurement site, e.g. Cologne, Düsseldorf, and Frankfurt (Fig. 2). These air masses are therefore influenced by regional emission sources and are referred in the following to belong to the regional transport group.

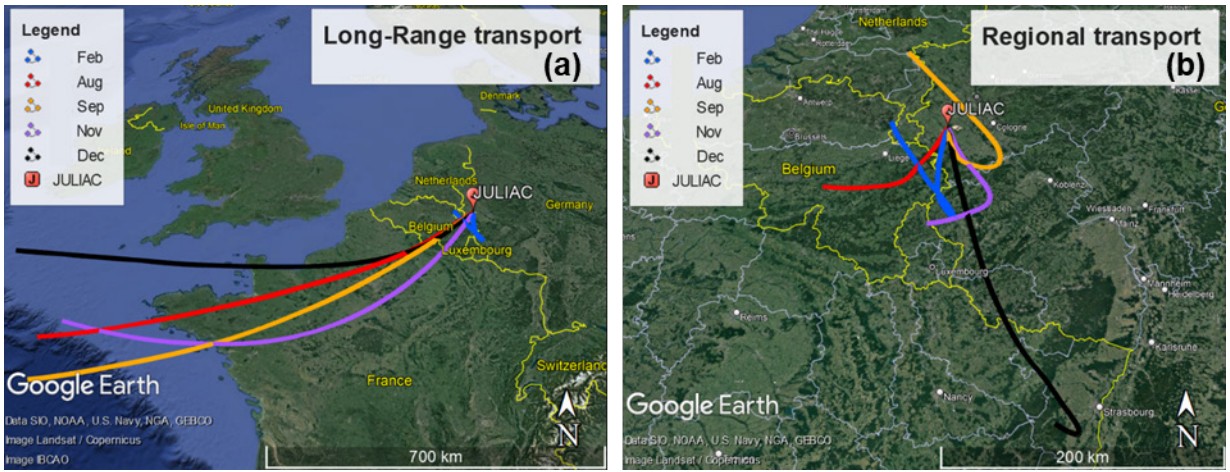

**Figure 2.** Results of the HYSPLIT cluster analysis of 48h back trajectories for the different measurement periods. (a) Trajectories from air masses originating from long-range transport for each period. (b) Trajectories from air masses from regional transport. © Google Maps 2022.

Fig. 3 shows mean diurnal profiles of $ClNO_2$, $NO_2$, $O_3$ concentrations, and photolysis frequencies of $ClNO_2$ in February, August, September, November, and December 2019, if measurements are split into 2 groups depending on the type of back trajectory associated to the measurement at that time. The complete

time series of measurements used for the analysis in this work are shown in Fig. S3-S7 (Supporting Information).

In all cases, the diurnal profiles of $ClNO_2$ showed an increase of concentration after sunset as can be expected from its chemical production during the night. Maximum concentrations were reached around midnight and $ClNO_2$ concentrations remained relatively constant until sunrise, when they started to decrease due to its photolysis.

The reaction chain to produce $ClNO_2$ at the night starts with the reaction of $NO_2$ and $O_3$. The median observed $O_3$ showed little diurnal variation in the cold seasons (February, November, and December) (Fig. 3). At this time of the year, the $O_3$ level was generally higher in long-range transported air (30 - 40 ppbv $O_3$) compared to regionally influenced air (15 – 20 ppbv $O_3$), for which ozone depletion by urban NO emissions was likely more important due to fresh emissions. During summer when photochemistry was most active (August, September), the median $O_3$ concentrations were considerably higher in regionally influenced air. Ozone mixing ratios in summer showed distinct diurnal profiles with noontime maxima of 80 ppbv in August and 40 ppbv in September, and nighttime values between 20 and 30 ppbv. In contrast, long-range transported air exhibited a less pronounced diurnal variation in the $O_3$ concentration and mixing ratios were often only between 20 and 40 ppbv. The high summertime ozone concentrations in regionally-transported air is likely due to fresh emissions of NO and VOCs, which are photochemically converted to $O_3$.

The influence of fresh emissions from nearby sources is also visible in the measured $NO_2$ concentrations, which were higher in regional air masses compared to concentrations in long-range transport air masses during the entire year. For regionally-transported air masses, average nocturnal $NO_2$ mixing ratios were around 10 ppbv in all measurement periods, except in December, when mixing ratios were lower with values of about 5 ppbv. In the night, median $NO_2$ concentrations in long-range transported air masses were generally lower than 5 ppbv in all seasons.

The age of the airmass could play a role in the observed levels of $ClNO_2$ due to the impact on $NO_2$ and $O_3$ concentrations, and hence on $ClNO_2$. As shown in Fig. 2, regionally transported air masses spend more time over urban areas picking up anthropogenic emissions (indicated by high $NO_2$ mixing ratios). They also have more time for the photochemical processing of pollutants compared to the long-range transported air masses. In the cold months (February, November, and December), long reaction times would lead to lower $O_3$ concentrations for the regional air masses due to the titration by anthropogenically emitted NO compared to conditions in August and September when photochemical ozone production is more efficient than the titration effect.

The nocturnal ClNO$_2$ concentrations were consistently lower in air masses from long-range transported air compared to regional transported air in nearly all seasons except again in December. The maximum median nighttime values were around 0.2 ppbv in long-range transported air and around 0.5 ppbv in air masses from regional transport (Fig. 3). Only in December, no significant dependence of the ClNO$_2$ concentration on the origin of air masses was observed.

Maximum ClNO$_2$ mixing ratios of 1.6 ppbv (15-min average), which were observed at 03:00 UTC on September 15 in the JULIAC campaign (Fig. S5, Supporting Information), are comparable to observations in other field campaigns. In Europe, high ClNO$_2$ mixing ratios have also been observed during summer in several field campaigns, in which ClNO$_2$ was measured: 0.8 ppbv near Frankfurt, Germany (Phillips et al., 2012), 0.8 ppbv in London, UK (Bannan et al., 2015) and 1.1 ppbv in Weybourne, UK (180 km northeast of London, (Sommariva et al., 2018)).

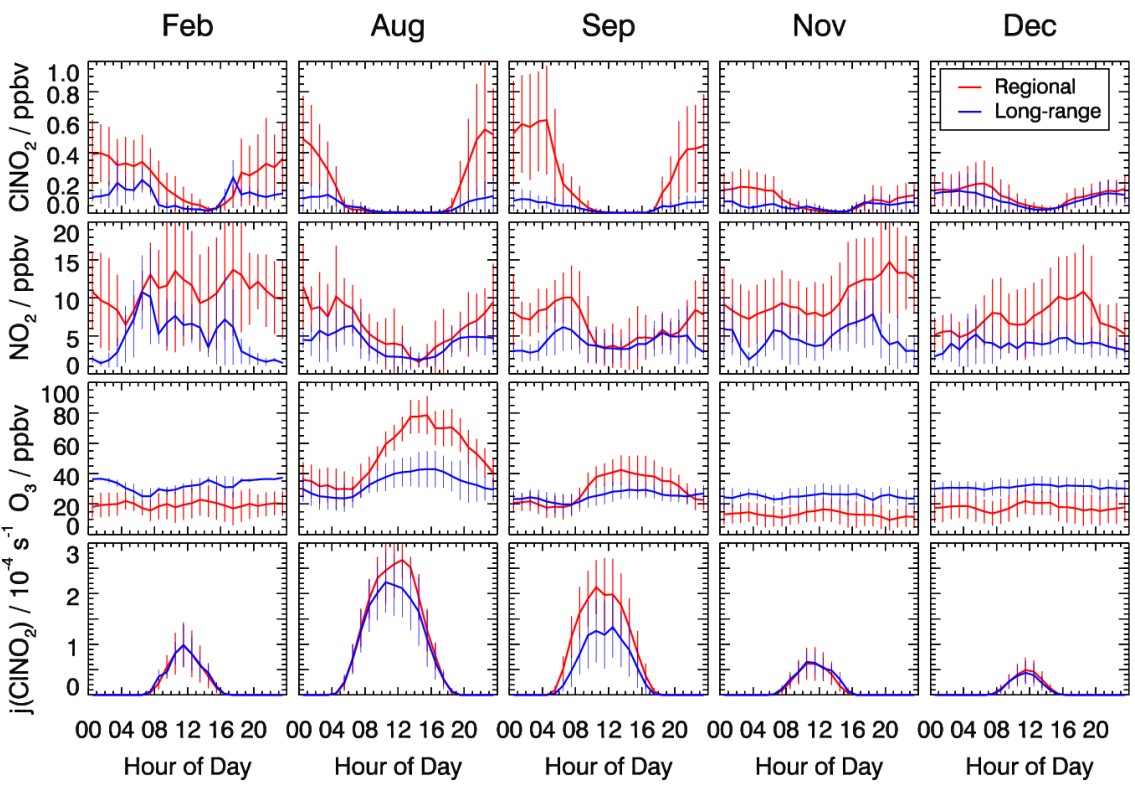

**Figure 3.** Mean diurnal profiles of ClNO$_2$, NO$_2$, and O$_3$ concentrations, and ClNO$_2$ photolysis frequencies. Trace gas concentrations were measured in the inflowing air or values measured inside the chamber were used to back-calculate concentrations in the inflowing air. Data are 1-h average values with error bars denoting 1σ standard deviations.

The seasonally varying photolysis frequencies of $ClNO_2$ showed diurnal noontime maxima of $0.4\times10^{-4}$ $s^{-1}$
in winter and $2.5\times10^{-4}$ $s^{-1}$ in summer. Sunlight lasted longest in summer and photolysis frequencies were
sufficiently high to destroy all $ClNO_2$ before midday. In contrast, daytime $ClNO_2$ concentrations remained
significantly above zero (around 30 pptv) in the cold seasons, because the maximum photolysis
frequencies were a least a factor of 2 lower than in summer and the duration of daylight was not long
enough to deplete all $ClNO_2$. Similar results were observed in wintertime measurements of $ClNO_2$ by
Sommariva et al. (2021).
Seasonal differences in $ClNO_2$ concentrations observations in this work can be compared to the seasonal
variations reported for measurements performed in Leicester, UK (Sommariva et al., 2018). In Leicester,
the highest $ClNO_2$ mixing ratio of 0.73 ppbv was observed in February, when also $NO_2$ mixing ratios
were highest with values of 43 ppbv. The seasonality of $ClNO_2$, $NO_2$, and $O_3$ observed during the
JULIAC campaign was different from the seasonality observed in Leicester. In this work, the highest
$ClNO_2$ concentrations were experienced in summer, when the air was influenced by emissions from
nearby cities (regional transport) resulting in high $NO_2$ and $O_3$ concentrations. The different seasonal
behavior in Jülich and Leicester suggests that the controlling factor for the production of $ClNO_2$ could
have been different in the two locations (Section 3.5).

## 3.3 Influence of the nocturnal vertical stratification of air on $ClNO_2$ concentrations

The $ClNO_2$ measurements presented in this work were obtained in air sampled at a height of 50 m above
ground (Section 2). While there is a well-mixed layer due to convection during the day, the cooling of the
ground results in weak convection of air after sunset leading to stratification of the air in the night.
In general, layers can be identified by the vertical profile of the potential temperature. In the night, a
stable surface layer (typically <20 m height) is expected to be formed, in which emissions from the
ground are trapped.  A weakly stable nocturnal boundary layer is on top of the surface layer (NBL,
typically in the height range between 20 m and 200 m) and a residual layer that is fully decoupled from
the ground (typically height >200 m) (Brown et al., 2007). Because the tip of the inlet of the SAPHIR-
JULIAC inlet system was 50 m above the ground, it was most often located within the nocturnal
boundary layer and thus impact of surface emissions in the sampled air is expected to be small.
This was particularly the case in the cold seasons (Feb., Nov., and Dec.) suggesting that most of the
nighttime measurements presented in this work are representative of conditions in the NBL. Similar
conditions were encountered in the summer during nights with low wind speed and cloudless conditions.
However, in 8 out of 30 nights from 20 August to 20 September, the sampled air at 50 m height was
temporarily influenced by surface air. Indicators were, for example, observed enhancements of the NO
and CO concentrations, and reduced mixing ratios of $ClNO_2$.
An example of such an event is shown in Fig. 4 which presents measurements from the night of 21 to 22
August 2019. After sunset (around 19:00 UTC), a stable surface layer was formed as indicated by a
positive vertical temperature gradient in the lowest 20 m (Fig. 4a). Until 22:00 UTC, the surface layer
height increased and developed a strong temperature inversion at 30 m height. Above the surface layer,
the temperature gradient was slightly positive up to a height of 80 m. It is expected that for the conditions
until about 22:00 UTC, the measured air at 50 m height was not influenced by surface emissions. During
this time, $ClNO_2$ mixing ratios increased continuously to 1.5 ppbv due to chemical production. After
22:30 UTC, $ClNO_2$ decreased to 0.5 ppbv until 0:00 UTC. The decrease coincided with an increase in
wind speed from below 2 m/s at 23:00 UTC to about 4 m/s at 0:00 UTC. This might be related to the
phenomenon of "nocturnal jets" that can produce high wind speeds at low altitudes already in a range of
50 m. The elevated wind speed and change of wind direction indicate air mass came down the Rur valley
from Düren, a small city 10 km away from the site. At the same time, the steep temperature gradient of
the inversion at 30 m disappeared and most likely facilitated entrainment of surface air with lower $ClNO_2$
concentration. This assumption is supported by an enhanced NO mixing ratio of 0.2 ppbv observed
shortly before midnight, indicating ground emissions (Fig. 4(b)). At the same time, the $NO_2$ mixing ratio
increased and the $O_3$ mixing ratio decreased by a similar amount (10 ppbv) likely due to the chemical
titration of $O_3$ by freshly emitted NO (Fig. 4(c)). The drop of $ClNO_2$ may have been caused by the lower
$ClNO_2$ production in the surface layer, because $N_2O_5$ concentrations were low due to $N_2O_5$ and $NO_3$ loss
on surfaces and chemical loss in reactions with NO and organic compounds that have emission sources on
the ground. At later times in this night, $ClNO_2$ mixing ratios increased again to a value of 1.3 ppbv at
01:00 UTC (Fig. 4(b)), when the air was again sampled from within the nocturnal boundary layer, where
loss processes are expected to be smaller compared to the surface layer.

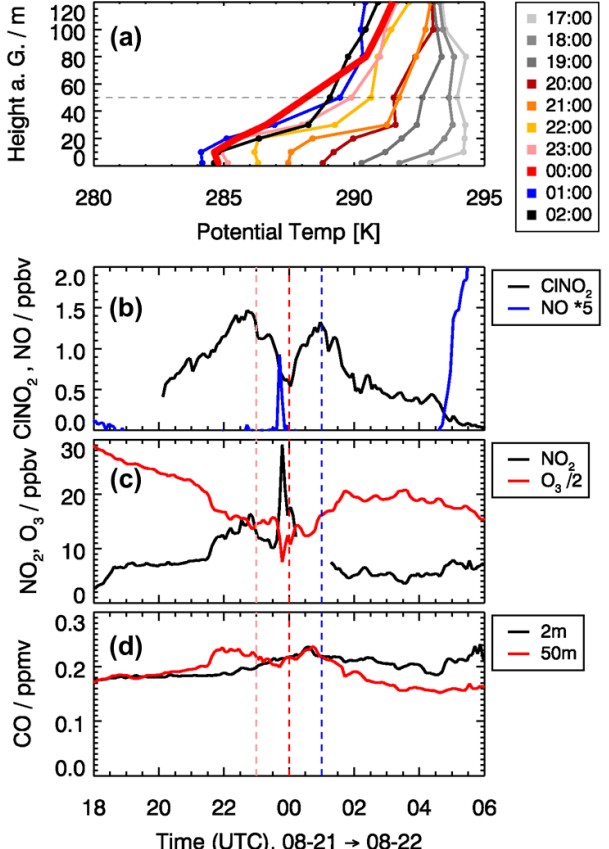


**Figure 4.** Impact of the vertical structure of air masses during the night from 21 to 22 August on observed trace gas concentrations. (a): Vertical profiles of the potential temperature derived from temperature measurements at different heights (2 m, 10 m, 20 m, 30 m, 50 m, 80 m, 100 m, 120 m). (b)-(d): $ClNO_2$, NO, $NO_2$, $O_3$, and CO mixing ratios sampled at 50 m height with the JULIAC-SAPHIR inlet system. CO mixing ratios were also measured at a height of 2 m. Colors of vertical lines correspond to colors of the vertical profiles of the potential temperature.

The median diurnal profiles presented in Section 3.2 include all measurements. The different behavior observed during the night, when air was temporarily impacted by surface interaction only constitute a small fraction of the measurement time. To quantify the influence of surface interactions, elevated NO concentrations at the sampling point can be used. For more than 90% of the time, measured NO mixing ratios are lower than 0.1 ppbv (Fig. S8, Supporting Information) indicating that air masses were typically little influenced by surface emissions. Therefore, it can be assumed that the sampling point was most often located in the nocturnal boundary layer. Median values further analyzed in this work are representative of conditions in the nocturnal boundary layer.

## 3.4 ClNO$_2$ production efficiency

The ClNO$_2$ production efficiency ($\varepsilon$) defined in Eq. 4 is affected by (1) the thermal equilibrium between NO$_3$ and N$_2$O$_5$, (2) the loss of NO$_3$ + N$_2$O$_5$ by reaction of NO$_3$ with VOCs and heterogeneous uptake of N$_2$O$_5$ on the aerosol surface, and (3) the yield of ClNO$_2$ from the heterogeneous reaction of N$_2$O$_5$. The value of the production efficiency cannot be simply calculated because the required parameters along the trajectory of the studied air mass are not known. Instead, a mean value of $\varepsilon$ is estimated empirically from the observed nocturnal increase of the ClNO$_2$ concentration at the measurement site and the corresponding integrated NO$_3$ production rate. This approach assumes that there are no significant nocturnal ClNO$_2$ losses in the studied air.

$$\varepsilon_t = \frac{([ClNO_2]_t - [ClNO_2]_{t0})}{\int_{t0}^{t} P(NO_3)_{(t)} dt} \qquad \text{(Eq. 6)}$$

For the calculation of the efficiency (Eq. 6) from measured ClNO$_2$ concentrations, the ClNO$_2$ concentration at sunset ($[ClNO_2]_{t0}$) is subtracted, because this fraction of ClNO$_2$ can be assumed to be produced in the previous night. This correction is important, especially for conditions in winter and late autumn, when tens of pptv of ClNO$_2$ were observed before sunset because of the long chemical lifetime of ClNO$_2$ under these conditions (Fig. 3).

An accurate calculation of the integrated NO$_3$ production rate would require the knowledge of the NO$_2$ and O$_3$ concentrations while the air mass is being transported, but the exact concentrations are only known at the location of the JULIAC tower. Therefore, it is necessary to make assumptions about the history of the air mass. For simplification, it is here assumed that the airmass arriving at the JULIAC site is homogeneous along the trajectory after sunset. This assumption requires that the consumption of NO$_2$ by reaction with O$_3$ is small over the integration time and that the chemical composition of the studied air remains undisturbed by mixing with air masses containing different trace gas concentrations. The latter assumption seems reasonable when the air is sampled above the nocturnal surface layer which was largely the case during the JULIAC campaign (Section 3.3). For these assumptions, the integrated NO$_3$ radical production P(NO$_3$) can be calculated from the measured NO$_2$ and O$_3$ concentrations at the measurement site and the reaction rate constant ($k_2$) of their reaction. The value of the reaction rate constant is taken from recommendations by IUPAC (Atkinson et al., 2004). Therefore, the production rate of NO$_3$ radical can be substituted by the reaction rate of NO$_2$ and O$_3$ and Eq. 6 is rewritten as following:

$$\varepsilon_t = \frac{[ClNO_2]_t - [ClNO_2]_{t0}}{\int_{t0}^{t} k_2 [NO_2]_t [O_3]_t dt} \qquad \text{(Eq. 7)}$$

$t_0$ can be set to the time of sunset and the time $t$ is stepwise increased by intervals of 5 minutes (time
resolution of the dataset) to calculate the time series of the production efficiency in one night. For further
analysis, the first 4 hours after sunset is averaged for each night, because $ClNO_2$ increased to its
maximum concentration in most of the nights of this campaign during this time. This suggests chloride is
not a limiting factor for $ClNO_2$ production. Mean values of the $ClNO_2$ production efficiency in each
season can then be compared.
The $ClNO_2$ production efficiency does not show a clear seasonal behavior, but values are larger in
regional transported air masses than in long-range transported air masses (Fig. 5). Mean values exhibit a
similar pattern, if values are taken from the entire night or a period in the second half of the night (Fig.
S9, Supporting Information).

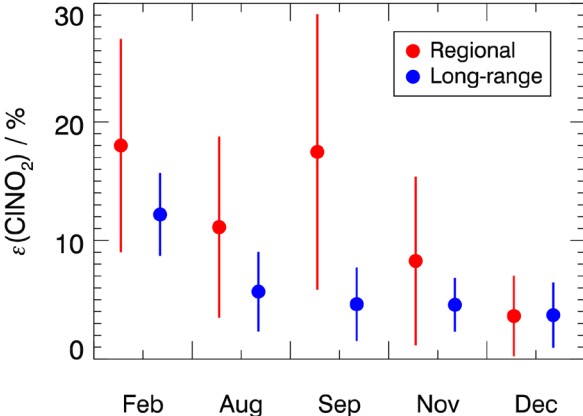


**Figure 5.** Mean $ClNO_2$ production efficiency for each measurement period for 4-hour average values
starting after sunset. Values are calculated for air masses originating either from regional or long-range
transportation. The vertical bars denote $1\sigma$ standard deviations.
For the air masses from regional transportation, the highest mean $ClNO_2$ production efficiency of
$(18\pm9)$ % was observed in February. This is consistent with a high $NO_3$ production rate due to high $NO_2$
concentrations (Fig. 3), as well as the low temperatures in February which favor the formation of $N_2O_5$.
Similar $ClNO_2$ production efficiency was observed in September, although $NO_2$ concentrations were low.
This suggests that other factors, besides the ones included in Eq. 4, contributed to the efficient production
of $ClNO_2$ in regional air masses in September.
The $ClNO_2$ production efficiencies obtained in December are similar with values of $(3\pm3)$ % for both
regional and long-range transportation air masses. This is consistent with observations of $ClNO_2$, $NO_2$,
and $O_3$ concentrations, which were also similar regardless of the origin of air masses in December (Fig.
3). In the other seasons, however, the $ClNO_2$ production efficiencies were 30 to 50% lower in air masses
from long-range transportation compared to values obtained for regional air masses. This can be
explained by elevated $NO_2$ concentrations in regional air masses, which shifts the equilibrium between
$NO_3$ and $N_2O_5$ to the side of $N_2O_5$ and $N_2O_5$ and therefore facilitates the production of $ClNO_2$.
It should be mentioned that the production of $ClNO_2$ also requires the availability of particulate chloride
(Reaction R4). During the JULIAC campaign, particulate chloride concentrations were measured by an
aerosol mass spectrometry (AMS) instrument giving average concentrations of (0.15±0.08), (0.07±0.03),
(0.07±0.06), and (0.09±0.04) $\mu g/m^3$ for measurements in February, August, September and November,
respectively (measurements in December were not available, see Table S2 in the Supporting Information).
The particulate chloride measurements by the AMS instrument are restricted to non-sea-salt aerosol,
because the AMS was operated to measure the non-refractory particulate matters. As the measurement
site is only 200 km away from the North Sea, sea salt was likely an important source of chloride in the
JULIAC campaign. Thus, there was most likely more chlorine present than measured by the AMS and the
observed chloride concentrations must be regarded as a low limit. Nevertheless, the high $ClNO_2$
production efficiency in the regional air masses suggests that particulate chloride was not a limiting factor
for the formation of $ClNO_2$ at the measurement site (for the period of 4-hours since sunset). In the
following analysis, it is assumed that the availability of particulate chloride was enough to sustain
reaction R4 during this study, so $ClNO_2$ production was only dependent on the availability of its gas
phase precursors (see Section 3.5).
Previous studies have reported similar values of $ClNO_2$ production efficiencies. Two field studies
performed in urban environments in Canada found median values of the $ClNO_2$ production efficiency of
1.0 % (Mielke et al., 2016) and 0.17% (Osthoff et al., 2018). These low values were attributed to gas-
phase loss reactions of $NO_3$ competing with the formation of $ClNO_2$. In addition, the authors determined
significant $O_3$ destruction by deposition and titration in the reaction with NO in a shallow nocturnal
boundary layer, which further limited the production of $ClNO_2$ (Osthoff et al., 2018). In another
campaign, measurements were performed on board of a ship during a cruise in the Mediterranean Sea
(Eger et al., 2019). The $ClNO_2$ production efficiency determined from these measurements was in the
range between 1% and 5%, attributed to the efficient gas-phase loss of $NO_3$ and to the high temperature
(usually >25°C) that shifted the thermal equilibrium towards $NO_3$, so that little $N_2O_5$ was expected. In
contrast, the $ClNO_2$ production efficiency observed in Pasadena, U.S. (Mielke et al., 2013) was much
higher than in the field studies in Canada (median value: 9.5%). These measurements were performed in
the coastal boundary layer, which was characterized by high concentrations of pollutants. The authors
attributed the high $ClNO_2$ production efficiency to the rapid $N_2O_5$ reaction with Cl that was present in
submicron aerosol particles from the redistribution of sea salt chloride as proposed by Osthoff et al.

538    (2008).

## 3.5 Dependence of the ClNO$_2$ production on the availability of NO$_2$ and O$_3$

Most of the measurements taken during nighttime from a height of 50 m were not affected by fresh local
emissions from the ground surface, as discussed in Section 3.2. As first approximation, it can be assumed
that particulate chloride is not limiting the formation of ClNO$_2$ (Section 3.4). Therefore, the amount of
ClNO$_2$ that can be formed during the night is a function of the amounts of NO$_2$ and O$_3$ available at sunset.
The dependence of the ClNO$_2$ production on the availability of NO$_2$ and O$_3$ for ambient conditions is
further investigated by box model calculations. This method was previously used by Sommariva et al.
(2018) and a detailed description can be found in their work. In brief, the model is initialized with a
matrix of initial NO$_2$ and O$_3$ concentrations. The chemical box model includes production and loss
reactions of ClNO$_2$ (Reaction R1- R4, reactions rate constants are taken from the IUPAC
recommendations (Atkinson et al., 2004)). ClNO$_2$ concentrations are calculated for each initial NO$_2$ and
O$_3$ concentrations after 4 hours. This length of the simulation is chosen, because observed ClNO$_2$
concentrations typically reached their maximum values approximately 4 hours after sunset in the JULIAC
campaign.
In the model, the efficiency of the conversion of N$_2$O$_5$ to ClNO$_2$ is assumed to be constant, with a value
for the uptake coefficient of N$_2$O$_5$ of 0.01 from Bertram and Thornton (2009), and a ClNO$_2$ yield of 0.5
(Reaction R4) from Roberts et al. (2009). The aerosol surface area ($S_a$) measured during JULIAC was in
the order of 100 μm$^2$/cm$^3$ (Table S1) and was set to this constant value in the model. Temperature was
fixed at 22 °C to represent typical summer-like conditions. Hence, the pseudo-first order reaction rate
constant for N$_2$O$_5$ uptake is 6.0×10$^{-5}$ s$^{-1}$. Following Sommariva et al. (2018), a constant NO$_3$ loss rate is
used to represent the typical loss of NO$_3$ radicals ($k_{NO3}$) in their reactions with organic compounds
(Reaction R5). The assumed value of the NO$_3$ loss rate, $k_{NO3}$, is adjusted, so that the modelled ClNO$_2$
concentration agrees with the magnitude of the observations (Fig. S10, Supporting Information), which
corresponds to an NO$_3$ reactivity of 0.004 s$^{-1}$. It should be noted that the purpose of such simplified model
is to examine the idealized dependence of ClNO$_2$ on the chemical conditions, not to reproduce the
measurements.
Fig. 6(a) shows the modelled ClNO$_2$ mixing ratios as a function of the initial NO$_2$ and O$_3$ concentrations
at sunset. Given the chemical conditions of long-range transported air masses in summer (25 to 35 ppbv
O$_3$ and 4 to 5 ppbv of NO$_2$), the model predicts ClNO$_2$ mixing ratios in the range of 0.1 to 0.16 ppbv.
Because of the simplifications adopted in the modelling approach, calculated ClNO$_2$ mixing ratios tend to
underestimate the measurements the measurements, which are around 0.2-0.3 ppbv (Fig. S10). For
regional air masses containing higher $NO_2$ mixing ratios (6 to 10 ppbv of $NO_2$), $NO_3$ production rates and
therefore also calculated $ClNO_2$ mixing ratios are higher (between 0.2 and 0.4 ppbv, Fig. 6(a)). Given the
position of each measurement periods in the isopleth plot, it can be concluded that all long-range
transported air masses tend to be $NO_2$ limited while the regional transported air masses tend to be $NO_2$
limited in summer/autumn and $O_3$ limited in winter.
To further interpret the controlling factors of $ClNO_2$ production, the dependence of $ClNO_2$ production
efficiency $\varepsilon$ on $NO_2$ and $O_3$ is presented in Fig. 6 (c). The modelled $ClNO_2$ production efficiency
increases with increasing mixing ratios of $NO_2$ but not with increasing $O_3$ (Fig. 6 (c)) as expected from
Eq. 4, which shows that the $ClNO_2$ production efficiency is a function of multiple parameters but not of
the $O_3$ mixing ratio. In general, the model reproduces the experimentally determined $ClNO_2$ production
efficiency (as shown in Fig. 5) within the uncertainty of the calculation (30% to 40%), which is mainly
due to the assumptions concerning the history of air masses (Section 3.4). However, the relatively high
$ClNO_2$ production efficiency found in August and September in the regional air masses (Fig. 5) is
significantly underestimated by the model. The discrepancy suggests that other processes facilitate the
conversion from $NO_3$ to $ClNO_2$ in the regional air masses for summer-like conditions. Though the
purpose of this model calculation is not to reproduce the observations, it is critical to address the related
uncertainties/limitations due to the assumptions in the simplified model. The key parameters affecting the
formation of $ClNO_2$ concentrations are temperature, $NO_3$ loss, $N_2O_5$ loss. Their impact on the model
results is discussed below.
Fig. 6(b) shows the dependence of modelled $ClNO_2$ on the temperature and $NO_2$ concentrations
investigated by the same model approach, for which the $O_3$ concentration is fixed to 30 ppbv
(representing typical $O_3$ level of long-range transported air). In this case, the modelled $ClNO_2$
concentrations reach maximum values at temperatures of 5 °C. For these winter-like conditions, the low
temperature shifts the equilibrium between $NO_3$ to $N_2O_5$ to the side of $N_2O_5$. In contrast, the conversion of
$NO_2$ to $ClNO_2$ is suppressed at high temperatures (T > 15°C) as typical conditions in August and
September.  Temperature also plays an important role for the value of the $ClNO_2$ production efficiency
due to the shift of the equilibrium between $NO_3$ to $N_2O_5$. The significantly higher $ClNO_2$ production
efficiency observed in February compared to the other seasons could be largely attributed to the low
temperature at that time (Fig. 6(d)).
Sensitivity tests demonstrate that decreasing the rate of the chemical loss of $NO_3$ to organic compounds
(Fig. S11, Supporting Information) only have small impact, while the seasonal variation of chemical loss
of $NO_3$ peaks in summer-like conditions due to the intense biogenic emission. The higher production
efficiency could be attributed to faster than assumed conversion from $N_2O_5$ to $ClNO_2$, which can bring
modelled and measured valued into agreement. This can be either achieved by increasing the value of the
$N_2O_5$ uptake coefficient (Fig. S12, Supporting Information) or the yield of $ClNO_2$ in the process of the
heterogeneous uptake of $N_2O_5$ on aerosol (Fig. S13, Supporting Information).
As mentioned above, the $NO_3$ reactivity is assumed to be 0.004 $s^{-1}$ to match the observations, which is
comparable to the $NO_3$ reactivity observed at a mountainous site in south Germany with a campaign-
averaged value of 0.01 $s^{-1}$ for nighttime conditions (Liebmann et al., 2018). As shown in the sensitivity
test, a higher $NO_3$ reactivity leads to lower modelled $ClNO_2$ concentrations. Therefore, the low $NO_3$
reactivity in the model could be regarded as a lower limit given the similar biogenic-influenced
environments.
In this model calculation, the aerosol surface area $S_a$ is held constant instead of using the value measured
inside the chamber, which was likely impacted by the sampling system but cannot be corrected for
ambient measurement (Section 2.3). Nevertheless, the measured $S_a$ gives some confidence that the model
is not using an unrealistic lower limit.
The aerosol chemical composition also plays a role in determining the production efficiency. The yield of
$ClNO_2$ from $N_2O_5$ heterogenous reaction ($\varphi(ClNO_2)$) can be expressed by assuming that the production
of $ClNO_2$ results from the competition between $Cl^-$ and $H_2O$ reacting with the $H_2ONO_2^+$ intermediate
formed from the $N_2O_5$ uptake on aerosol (Bertram and Thornton, 2009;Mielke et al., 2013;McDuffie et
al., 2018b).
$$\varphi(ClNO_2)_{par} = \left(1 + \frac{[H_2O]}{50[Cl^-]}\right)^{-1} \qquad \text{(Eq. 8)}$$
The value of the $ClNO_2$ yield is different in the periods of the campaign showing maximum values of 0.6
to 0.8 in February (Fig. S14, Supporting Information). This is consistent with the relatively high $ClNO_2$
production efficiency derived from the integrated production rate of $NO_3$ (Eq.7). However, the calculated
$ClNO_2$ yield decreases below 0.4 in August and September, which could be attributed to the higher
aerosol liquid water content in these two periods compared to the value in other periods (Table S1). The
calculated $ClNO_2$ yield is also higher for the long-range transported air masses than those for the regional
one (Fig. S14, Supporting Information). The relatively high $ClNO_2$ production efficiencies found in the
regional air masses, which are in contrast to their relatively low calculated $\varphi(ClNO_2)$, suggest that other
factors play an important role in determining the $ClNO_2$ production such as larger-than-assumed uptake
coefficient for $N_2O_5$ and/or aerosol surface area.

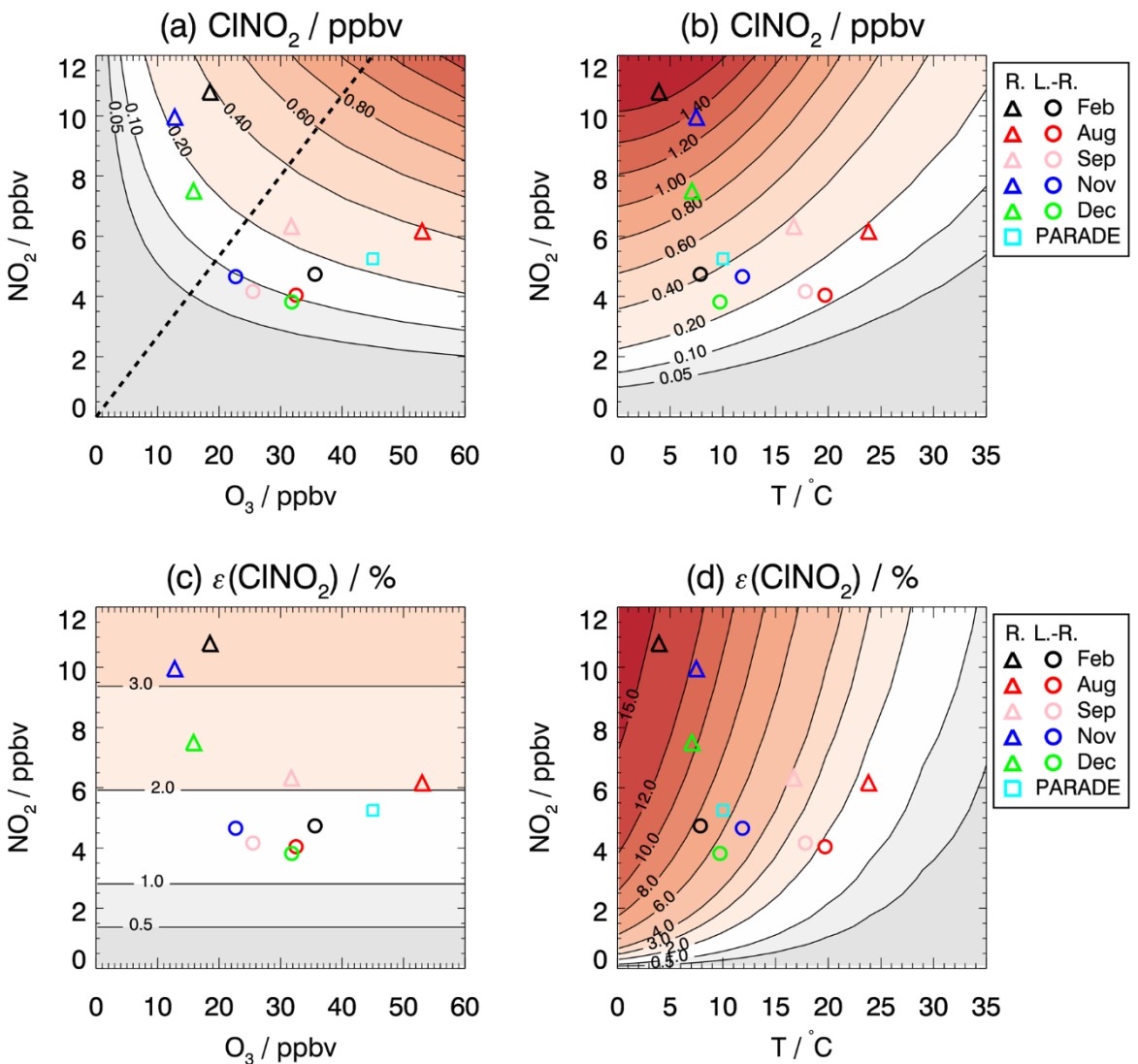


**Figure 6.** Isopleth plot of modelled (a, b) $ClNO_2$ mixing ratios that accumulate during nighttime and (c, d) the $ClNO_2$ production efficiency depending on (a, c) the initial $O_3$ and $NO_2$ mixing ratios and (b, d) the temperature and initial $NO_2$ mixing ratios. Values are taken after four hours since sunset, when maximum $ClNO_2$ concentrations were observed. Symbols mark calculated $ClNO_2$ mixing ratios for average values of $NO_2$ and $O_3$ mixing ratios measured in each period of the JULIAC campaign, if air masses originated either from long-range (L.-R.) or regional (R.) transportation. For comparison, values are also shown for measurements during the PARADE campaign in summer in Germany (Phillips et al., 2012). The dashed line (panel a) separates the regimes for which $ClNO_2$ production is more sensitive to the change of $O_3$ (upper left) and $NO_2$ (bottom right).

642

643 For comparison, the observation from another field campaign conducted in a similar rural environment in
644 Germany is marked in the isopleth diagram (Fig. 6). The PARADE campaign took place in the Taunus
645 Observatory of the University of Frankfurt located 170 km southeast of the JULIAC measurement site
646 (Phillips et al., 2012). The maximum observed $ClNO_2$ mixing ratio was 0.8 ppbv when the measurement
647 site was influenced by air masses from the UK/North Sea. This value is lower than the results of the
648 model calculations using the median $NO_2$ and $O_3$ observed in that campaign, which is consistent to the
649 general underprediction for summer-like conditions for the JULIAC campaign, suggesting the conversion
650 from $NO_3$ to $ClNO_2$ is more efficient than the model predicts in summer. The position in the isopleth
651 diagram suggests that $ClNO_2$ formation was limited by the availability of $NO_2$ similar to the summer
652 period the JULIAC campaign, the same season as the PARADE campaign (August).

## 4 Summary and Conclusions

654 Concentrations of $ClNO_2$ and other trace gases and the chemical composition of aerosols were measured
655 during the Jülich Atmospheric Chemistry Project (JULIAC) campaign in 2019 performed at a rural site in
656 Germany. Ambient air was sampled into the atmospheric simulation chamber SAPHIR from a height of
657 50 m, which, for most of the time, was uncoupled from surface layer during the night. Chamber
658 characterization experiments demonstrated that no significant loss or production of $ClNO_2$ occurred inside
659 the chamber for experimental conditions of the JULIAC campaign.

660 In all periods, $ClNO_2$ measurements showed a trend of increasing mixing ratios after sunset with
661 maximum values were reached around midnight. This qualitative behavior is consistent with the chemical
662 production of $ClNO_2$ and insignificant losses during the night. Photolysis was the main loss process for
663 $ClNO_2$ on the following day. The maximum $ClNO_2$ concentration in this campaign of 1.6 ppbv was
664 observed in September at the late-night (03:00 UTC). The analysis of the origin of air masses by
665 calculations of back trajectories shows that mixing ratios of $ClNO_2$, $NO_2$ and $O_3$ were higher in regional
666 air masses than in air masses that traveled a long distance.

667 A case study analyzing measurements in the night from 21 to 22 August 2019 shows that the stratification
668 of layers during the night can strongly impact observed trace gas concentrations, specifically when the
669 sampling point of the inlet system was located within a height range that was characterized by poor
670 vertical mixing of the air. During most times of the campaign, however, the sampling point was isolated
671 from the surface layer during the night. In this case, losses of trace gases to the surface and reactions with
672 fresh emissions on the ground, which would typically reduce $ClNO_2$ production, were not important.

The ClNO$_2$ production efficiency (i.e. the number of ClNO$_2$ molecules formed per produced NO$_3$
molecule) was higher for conditions in air masses from regional areas than from long-range
transportation, mostly due to the higher NO$_2$ mixing ratios. The minimum average value of the production
efficiency calculated for the individual measurement periods in the JULIAC campaign was 3%,
experienced in December for all air masses independent from their origin. This low value can be
attributed to the low NO$_2$ mixing ratios experienced in winter. For the air masses from long-range
transportation, the mean ClNO$_2$ production efficiencies were in the range of 3% to 6% in the period
between August to November but were as high as 12% in February, consistent with the seasonality of the
observed ClNO$_2$ concentrations. The highest mean ClNO$_2$ production efficiency was found in February,
when values reached (18±9) % and NO$_2$ concentrations were highest in the regional air masses. High
ClNO$_2$ production efficiency was also found in September, when NO$_2$ concentrations were low,
suggesting that other factors including the available aerosol surface area ($S_a$), the variability of the N$_2$O$_5$
uptake coefficient, and the yield of ClNO$_2$ in the heterogeneous reaction of N$_2$O$_5$ were favoring the
production of ClNO$_2$.
With the help of a simple box model of nighttime chemistry for the NO$_3$-N$_2$O$_5$-ClNO$_2$ system, the
dependence of ClNO$_2$ concentration on the availability of O$_3$ and NO$_2$ was investigated. The purpose of
such simplified model is to demonstrate the general feature of ClNO2 production versus chemical
conditions but not to compare with observations. The model results suggest that ClNO$_2$ production was
more sensitive to the availability of NO$_2$ than that of O$_3$, especially the air masses from long-range
transportation. The seasonal variability of ClNO$_2$ is less pronounced compared to the seasonal changes of
NO$_2$ and O$_3$ concentrations, because changes in the NO$_2$ and O$_3$ concentrations partly compensated for
each other. The simple model cannot predict the seasonal changes in the observed ClNO$_2$ mixing ratios.
This indicates that processes other than the NO$_3$ production rate significantly impacted the ClNO$_2$ mixing
ratios. Nevertheless, this simple model approach helps to understand the general features of the
dependence of ClNO$_2$ concentrations on the availability of NO$_2$ and O$_3$ in the JULIAC campaign.

**Data availability**
The data used in this study are available on the Jülich DATA platform (https://doi.org/10.26165/JUELICH-
DATA/XG6YGZ).
**Author contributions**
AH designed JULIAC campaign and organized it together with HF and FH. ZT and RS performed the
measurements of ClNO$_2$ and analyzed the data. ZT, RS, HF, and AH wrote the paper. All co-authors
contributed with data and commented and discussed the manuscript and contributed to the writing of the
manuscript.

**Competing interests**

Some authors are members of the editorial board of Atmospheric Chemistry and Physics. The authors
declare that they have no conflict of interest.

**Acknowledgements**

The authors thank the scientific team of JULIAC campaign for logistical support, the Chemistry
Workshop and Glassblower of the University of Leicester for technical support

**Financial Support.**

This project has been supported by the BMBF project ID-CLAR (grant no. 01DO17036) and BMBF project
PRACTICE (grant no. 01LP1929A), the European Research Council (ERC) and European Commission
(EC) under the European Union's Horizon 2020 research and innovation program (SARLEP grant
agreement no. 681529 and Eurochamp 2020 grant agreement no. 730997).

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
