# Peer review of "Seasonal variation of nitryl chloride and its"

_Atmospheric Chemistry and Physics, 2022_

## Author Comment (AC1)

**Response to Comments**

**Reviewer #1:**

This manuscript presents observations of nitryl chloride (CINO2), along with several precursor compounds, measured over several seasons at 50m above the surface at a rural site in western central Europe. CINO2 is an important source of the powerful atmospheric oxidant atomic chlorine and has been shown to demonstrate significant spatial variability due to its relatively complex production mechanism. Although many measurements have been made of CINO2 at surface locations, the reduced concentrations close to nitric oxide (NO) emissions mean it is likely to be more efficiently produced in the nocturnal residual layer, from where subsequent mixing will allow it to influence surface photochemistry on the following day. The observations presented here are predominantly from within the nocturnal residual layer, and thus represent a significant contribution to the growing body of data on mid-continental CINO2.

The authors present the  $CINO_2$  data, and use co-located measurements of  $CINO_2$ precursors to calculate a CINO2 production efficiency to compare across the measurement period. This is a useful parameter on which to focus, as the complex nature of CINO2 production results in significant variability, making it often difficult to constrain in models. The authors then use a chemical box-model to explore the effects of ozone  $(O_3)$ , nitrogen dioxide  $(NO_2)$  and temperature on CINO2 production across the experienced parameter space. This analysis is insightful; however, I feel the authors need to do more to demonstrate the sensitivity of their analysis and conclusions to other important parameters that control CINO2 production. In particular, the sensitivity to the loss rate of the nitrate radical (NO3) to reaction with volatile organic compounds (VOC) warrants a more detailed sensitivity analysis than that presented in Fig. S9. The correlation between measured particulate chloride and calculated CINO2 production efficiency should also be shown to support the argument made that this is not a limiting factor. Overall, the manuscript is well written and represents a valuable contribution to the field, and warrants publication in ACP once the following comments have been addressed.

**We thank the reviewer for the useful comments/suggestions. Please find below our answers and the related revisions (in blue) to the manuscript.**

 As particle surface area and chloride content are key factors in the production of CINO2, it would be useful for the reader if these data were presented somewhere in the paper or supplement and discussed in more detail. In section 3.4 the authors argue that CINO2 production efficiency is not limited by particle chloride content, but I feel this statement would be better supported if the particle data were shown. Multiple factors can influence both the uptake of  $N_2O_5$  to particles and the subsequent yield of CINO2, such as chloride molarity and liquid water content. The authors acknowledge that they do not have sufficient data to fully characterize the particle phase, however, more could be done to demonstrate the sensitivity of the system to these parameters (e.g., McDuffie et al. 2018).

**Answer**: We agree that the aerosol surface area, the liquid water content and chloride concentration are important parameters to determine the CINO2 production. However, aerosol measurements were only conducted inside the chamber, and could be significantly affected by the sampling system (blower) and by the large surface of the chamber. We included the measured aerosol surface area data and calculated the liquid water content using aerosol thermal dynamic model ISORROPIA2 in Table S1 in the Supplement. The aerosol chemical composition is shown in Table S2. We modified the discussion starting at line 591 to address this issue "In this model calculation, the aerosol surface area  $S_a$  is held constant instead of using the *value* measured inside the chamber, which was likely impacted by the sampling system but cannot be corrected for ambient measurement (Section 2.3). Nevertheless, the measured  $S_a$  gives some confidence that the model is not using an unrealistic lower limit.

The aerosol chemical composition also plays a role in determining the production efficiency. The yield of CINO2 from N2O5 heterogenous reaction ( $\phi$ (ClNO2)) can be expressed by assuming that the production of CINO2 results from the competition between Cl- and H2O reacting with the H2ONO2+ intermediate formed from the N2O5 uptake on aerosol (Bertram and Thornton, 2009;Mielke et al., 2013;McDuffie et al., 2018).

$$\varphi(\text{CINO}_2)_{\text{par}} = \left(1 + \frac{[\text{H}_2\text{O}]}{50[\text{Cl}^-]}\right)^{-1}$$
 (Eq. 8)

The value of the CINO2 yield is different in the periods of the campaign showing maximum values of 0.6 to 0.8 in February (Fig. S14). This is consistent with the relatively high CINO2 production efficiency derived from the integrated production rate of NO3 (Eq.7). However, the calculated CINO2 yield decreases below 0.4 in August and September, which could be attributed to the higher aerosol liquid water content in these two periods compared to the value in other periods (Table S1). The calculated CINO2 yield is also higher for the long-range transported air masses than those for the regional one (Fig. S14, Supporting Information). The relatively high CINO2 production efficiencies found in the regional air masses, which are in contrast to their relatively low calculated  $\phi(CINO_2)$ , suggest that other factors play an important role in determining the CINO2 production such as larger-than-assumed uptake coefficient for N2O5 and/or aerosol surface area."

We also added the calculated  $\varphi(CINO_2)$  in the supplement.

**Figure S14.** Scatter plot of calculated CINO2 yield ( $\varphi$ (CINO2)) versus the ratio between chloride and aerosol liquid water content. Parameterized values of  $\varphi$ (CINO2) are calculated following literatures recommendations (Bertram and Thornton, 2009;Mielke et al., 2013;McDuffie et al., 2018). Red and blue dots denote the average for the regional and long-range transported air masses, respectively. In September, the data for long-range transported air masses case is missing due to the lack of simultaneous aerosol chemical composition measurements.

2. As with comment 1, the conclusions of the work would be better supported if the sensitivity to the gas phase loss of NO3 to VOC reaction was demonstrated (beyond that shown in Fig. S9). In the modelling work presented in Sect. 3.5 the authors assume a constant NO3 reactive loss rate (kNO3) of 0.001 s-1. As this work is carried out across both summer and winter seasons, and due to the strong biogenic control of the kNO3, it seems unlikely that this constraint is valid. Observations of kNO3 at another site in Germany have shown kNO3 values approaching 0.3 s-1 (Liebmann et al. 2018). Although the authors do carry out a set of simulations with a value of kNO3 = 0.0005 s-1, a more thorough assessment of the model sensitivity to this parameter would better support the authors assumption that it plays a minor controlling role.

**Answer:** We acknowledge that the NO3 reactivity is one of the major uncertainties in our model calculation. As discussed in the response of comment 3, the NO3 reactivity used in the model has been increased to match the

observed CINO2. The comparison to the observation serves as a justification of the choice of  $k_{NO3}$ . Liebmann et al. (page 12049-12050) report "Campaign-averaged values were low (~0.01 s-1) during night-time but a factor of 10 larger ~ 0.1 s-1 at 14:00UTC (local 16:00)." They state "The elevated location of the Hohenpeissenberg observatory, located on a mountain top above the surrounding countryside, favored sampling from the residual layer and free troposphere at night-time. In the absence of turbulent exchange, the residual layer and free troposphere may become disconnected from the planetary boundary layer (PBL) and thus from ground-level emissions of reactive trace gases; the layers may thus contain low levels of biogenic trace gases as well as low(er) levels of NO2 and higher levels of ozone." In our paper we also argue that we measured most of the time in the NBL that was disconnected from ground emissions. Therefore, we used their reported nighttime values of 0.01 s-1 as a reference for comparison with JULIAC, as we assume that CINO2 is formed mainly at night.

We added a sentence in Line 538 "The assumed value of the NO3 loss rate,  $k_{NO3}$ , is adjusted, so that the modelled CINO2 concentration agrees with the magnitude of the observations (Fig. S10, Supporting Information), which corresponds to an NO3 reactivity of 0.004 s-1." We also added in Line 591 "Though the purpose of this model calculation is not to reproduce the observations, it is critical to address the related uncertainties/limitations due to the assumptions in the simplified model. The key parameters affecting the formation of CINO2 concentrations are temperature, NO3 loss, N2O5 loss. Their impact on the model results<del>ions</del> is discussed below.

•••

As mentioned above, the NO3 reactivity is assumed to be 0.004 s-1 to match the observations, which is comparable to the NO3 reactivity observed at a mountainous site in south Germany with a campaign-averaged value of 0.01 s-1 for nighttime conditions (Liebmann et al., 2018). As shown in the sensitivity test, a higher NO3 reactivity leads to lower modelled CINO2 concentrations. Therefore, the low NO3 reactivity in the model could be regarded as a lower limit given the similar biogenic-influenced environments."

3. It would be useful if the observations overlaid on the model isopleths in Fig. 6 (a) and (b) showed the observed CINO2 mixing ratios to compare with the model values. Although the purpose of the modelling is not to recreate the observations, rather to investigate the chemical sensitivities of the system, it would provide confidence in the model's ability to accurately represent the chemistry if the general observational trends were recreated.

**Answer:** We have to admit that it is very difficult to add the CINO2 observation data in the isopleths in Fig. 6. Instead, we extract the modelled results from the isopleth plots, which are compared with the observation data and added a new figure to the Supplement (Fig. S10). We agree that the comparison helps to provide confidence on the model's ability to represent the chemical conditions. In fact, with the help of this comparison, we realize that the original model used too small  $k_{NO3}$  and overpredicted the modelled CINO2 concentrations. In the revised manuscript, the  $k_{NO3}$  is increased from 0.001 to 0.004 to better reproduce the magnitude of CINO2 (see answer to previous question).

We added a sentence in Line 562 "Following Sommariva et al. (2018), a constant NO3 loss rate is used to represent the typical loss of NO3 radicals ( $k_{NO3}$ ) in their reactions with organic compounds (Reaction R5). The assumed value of the NO3 loss rate  $k_{NO3}$  is adjusted so that the modelled ClNO2 concentration agrees with the magnitude of the observations (Fig. S10, Supporting Information), which corresponds to an NO3 reactivity of 0.004 s-1."

**Figure S10.** Comparison between observed and modelled  $CINO_2$  for the regional (left) and long-range (right) transportation air masses. Model results are calculated as done in Fig. 6 in main text but measured  $O_3$  concentrations and temperature data are used as input.

**References**

- Bertram, T. H., and Thornton, J. A.: Toward a general parameterization of N2O5 reactivity on aqueous particles: the competing effects of particle liquid water, nitrate and chloride, Atmos. Chem. Phys., 9, 8351-8363, 10.5194/acp-9-8351-2009, 2009.
- Liebmann, J. M., Muller, J. B. A., Kubistin, D., Claude, A., Holla, R., Plass-Dülmer, C., Lelieveld, J., and Crowley, J. N.: Direct measurements of NO3 reactivity in and above the boundary layer of a mountaintop site: identification of reactive trace gases and comparison with OH reactivity, Atmos. Chem. Phys., 18, 12045-12059, 10.5194/acp-18-12045-2018, 2018.
- McDuffie, E. E., Fibiger, D. L., Dubé, W. P., Lopez Hilfiker, F., Lee, B. H., Jaeglé, L., Guo, H., Weber, R. J., Reeves, J. M., Weinheimer, A. J., Schroder, J. C., Campuzano-Jost, P., Jimenez, J. L., Dibb, J. E., Veres, P., Ebben, C., Sparks, T. L., Wooldridge, P. J., Cohen, R. C., Campos, T., Hall, S. R., Ullmann, K., Roberts, J. M., Thornton, J. A., and Brown, S. S.: CINO2 Yields From Aircraft Measurements During the 2015 WINTER Campaign and Critical Evaluation of the Current Parameterization, J. Geophys. Res., 123, 12,994-913,015, 10.1029/2018jd029358, 2018.
- Mielke, L. H., Stutz, J., Tsai, C., Hurlock, S. C., Roberts, J. M., Veres, P. R., Froyd, K. D., Hayes, P. L., Cubison, M. J., Jimenez, J. L., Washenfelder, R. A., Young, C. J., Gilman, J. B., de Gouw, J. A., Flynn, J. H., Grossberg, N., Lefer, B. L., Liu, J., Weber, R. J., and Osthoff, H. D.: Heterogeneous formation of nitryl chloride and its role as a nocturnal NOx reservoir species during CalNex-LA 2010, J. Geophys. Res., 118, 10,638-610,652, 10.1002/jgrd.50783, 2013.
- Sommariva, R., Hollis, L. D. J., Sherwen, T., Baker, A. R., Ball, S. M., Bandy, B. J., Bell, T. G., Chowdhury, M. N., Cordell, R. L., Evans, M. J., Lee, J. D., Reed, C., Reeves, C. E., Roberts, J. M., Yang, M., and Monks, P. S.: Seasonal and geographical variability of nitryl chloride and its precursors in Northern Europe, Atmos. Sci. Lett., 19, e844, 10.1002/asl.844, 2018.

---

## Author Comment (AC2)

**Response to Comments**

**Reviewer #2:**

Tan et al. report the measurements of  $CINO_2$ ,  $NO_2$ ,  $O_3$  and related parameters for three seasons in 2019, obtained during the Jülich Atmospheric Chemistry Project (JULIAC) campaign in Germany. An important result of this study is the variations of  $CINO_2$  production efficiency in different seasons, which are most sensitive to the availability of  $NO_2$  and increase with the decreasing temperature. This finding is valuable as it enhances our understanding on the dependence of  $CINO_2$  formation on the availability of  $NO_2$  and  $O_3$  in Europe. Overall, the manuscript is well presented, however, I feel that the importance of the study and discussion of results can be further strengthen and improved. My comments are as below.

We thank the reviewer for the useful comments/suggestions. Please, find below our answers and the related revisions (in blue) to the manuscript

1. Line 19: Delete the word 'ion' (same for line 52)

Answer: Done.

2. Line 22: Please specify the date instead of using 'one night in September'

Answer: We specify it as "...in the night of September 20."

3. Line 58: The yield for CINO2 ( $\varphi_{ClNO_2}$ ) can be equal to 0 or 1, therefore, it should be  $\leq$

Answer: Corrected

4. Line 62–63: 'The forward and back reactions constitute a fast thermal equilibrium between NO3 and N2O5 that is established within a minute at room temperature.' Revise this sentence by justifying how the equilibrium can be establish within a minute. Is this base on the authors' calculation or from the literature? The concentration of NO2 can also influence the equilibrium of NO3 and N2O5

**Answer:** We revised the sentence as "The forward and back reactions constitute a fast thermal equilibrium between  $NO_3$  and  $N_2O_5$  that is quickly established at temperatures typically found in the lower troposphere (Brown and Stutz, 2012)."

5. Line 75: CINO2 usually present at night but not always is the case. Suggest to delete the word 'only'

**Answer:** The word "only" here refers to  $N_2O_5$  not to CINO2. We revised the sentence as "Therefore, significant concentrations of  $N_2O_5$  (the precursor of CINO2) are usually only present at night."

6. Line 123: The authors should highlight in the introduction or conclusion why investigation on the seasonal variation of CINO2 concentrations and its formation are scientifically important

**Answer:** We revised the last paragraph of the introduction as "In this work, the seasonal variation of CINO2 concentrations and its formation are investigated. As mentioned above, previous studies have demonstrated that CINO2 concentrations show significant seasonal variations (Mielke et al., 2016;Sommariva et al., 2018). However, intensive seasonal measurements in central Europe, to our knowledge, have not been performed so far. Given the ubiquitous nature of CINO2 and its importance to enhance atmospheric oxidation processes, more detailed studies are needed to broaden our knowledge of atmospheric CINO2 levels, its seasonal behavior and its distribution in environments with different chemical conditions. In addition, this work presents empirical production efficiencies of CINO2 determined from the nighttime measurements of CINO2, NO2 and O3 and analyzed for their seasonal variations and origin of air masses, a prerequisite to understand the contribution of CINO2 to radical photochemistry under the chemical and meteorological conditions encountered in this campaign. Finally, a chemical box model is used here to understand the dependence of CINO2 formation and production efficiency on the observed nocturnal  $O_3$  and  $NO_2$  concentrations. The measurements and analysis presented in this paper help to illustrate the seasonal variability of CINO2 concentrations and shed light on the factors that control its production in different seasons."

7. Line 168: The concentration of Cl2 in the cylinder used for calibration is 5 ppmv (±5%). As we know Cl2 is a very reactive gas that can loss on surfaces. Is this ± 5% a reliable value? The authors should provide details on whether they have quantified the output concentration of Cl2 from the cylinder and/or consider the potential loss of Cl2 in the calibration system, e.g. the loss on the regulator of the cylinder or tubing? This is crucial for the determination of CINO2 calibration factor

and estimation of measurement uncertainty, which can affect the reported levels of  $CINO_2$  and maybe the conclusions of this study.

**Answer:** As explained from Line 169 to 173, the CINO2 concentration in the calibration air is determined by measuring the NO2 concentration after thermally decomposing CINO2 to Cl and NO2. Thus, the absolute concentration of Cl2 does not influence the production of CINO2 or the accuracy of the calibration. As long as there is enough and stable Cl2 supply to the calibration unit, certain amount of CINO2 is produced. The presence of Cl2 is confirmed by the CIMS. The sentence about the uncertainty of Cl2 is misleading, and we deleted the sentence.

8. Line 279: Specify the humidity (RH) of the humidified chamber air

Answer: Done.

 Line 300: 'no corrections are needed for the interpretation of CINO2 measurements'. I am wondering if this variation has been considered in the estimation of the measurement uncertainty.

**Answer:** It was not considered in the measurement uncertainty given its low influence on the measurement (<1%).

10. Line 367: As shown in this figure, the CINO2 and related parameters are separated into long-range transport and region transport. The classification of long-range and regional has been described in the text. A lacking information here is the 'age' of different air masses. My question is will the 'age' of air masses play important role in the observed levels of CINO2? I think this should also be addressed in the discussion since the 'age' of air mass may affect the NO2 and O3 concentrations

**Answer:** We have added the following sentence to address the potential difference of 'age' between the two cases: "The age of the airmass could play a role in the observed levels of CINO2 due to the impact on NO2 and O3 concentrations, and hence on CINO2. As shown in Fig. 2, regionally transported air masses spend more time over urban areas picking up anthropogenic emissions (indicated by high NO2 mixing ratios). They also have more time for the photochemical processing of pollutants compared to the long-range transported air masses. In the cold months (February, November, and December), long reaction times would lead to lower O3 concentrations for the regional air masses

due to the titration by anthropogenically emitted NO compared to conditions in August and September when photochemical ozone production is more efficient than the titration effect."

11. Line 387: Section 3.3 describes the nocturnal vertical stratification and summarize that the JULIAC inlet (50 m) is most often located within the nocturnal boundary layer and on top of the surface layer. What does it mean by most often? At this point, I am not so convinced yet that the CINO2 are often measured above the nocturnal boundary layer with the discussion and provide only one day example (Fig.4). Please provide more evidence (of different seasons) and discussion in the main text or supporting info to support this argument. This is an important information for the validity of the calculation made from Eq7 (Line 465)

**Answer:** As NO can be regarded as an indicator of surface interruption, we added a plot Fig. S8 to show the cumulative frequency of measured NO concentrations to indicate the influence from surface interaction. We also added discussion at the end of section 3.3 "...when air was temporarily impacted by surface interaction only constitute a small fraction of the measurement time. To quantify the influence of surface interactions, elevated NO concentrations at the sampling point can be used. For more than 90% of the time, measured NO mixing ratios are lower than 0.1 ppbv (Fig. S8, Supporting Information) indicating that air masses were typically little influenced by surface emissions. Therefore, it can be assumed that the sampling point was most often located in the nocturnal boundary layer. Median values further analyzed in this work are therefore representative of conditions in the nocturnal boundary layer."

**Figure S8.** Cumulative histogram of measured NO concentrations during nighttime for different periods. The horizontal lines denote the position of 90% percentile of data.

12. Line 541–542: The measured aerosol surface area is an essential parameter for the calculation. This should be included in the supporting info. Can the authors justify why setting the aerosol surface area to constant value in the model since they have measurement data?

**Answer:** We agree that the aerosol surface area is an important parameter to calculation the CINO2 production. However, the measurement was only conducted inside the chamber, which could be significantly changed by the sampling system (blower). We included the measured aerosol surface area data in Table S1 in the Supplement to show the order of magnitude of this parameter and added the following sentence at line 591 to address this issue "In this model calculation, the aerosol surface area  $S_a$  is held constant instead of using the *value* measured inside the chamber, which was likely impacted by the sampling system but cannot be corrected for ambient measurement (Section 2.3). Nevertheless, the measured  $S_a$  gives some confidence that the model is not using an unrealistic lower limit."

13. Line 577–578: Temperature also plays an important role for the value of the  $CINO_2$  production efficiency due to the shift of the equilibrium between  $NO_3$  to  $N_2O_5$ . The temperature shift may also affect the humidity which has been shown

in previous studies to promote  $N_2O_5$  uptake and production of CINO2. How can the authors separate the effect of humidity with the effect of temperature?

**Answer:** In the simplified model, it's difficult to separate the effect between temperature and humidity. Instead, the ambient water content concentration is held constant, which means the RH increase with higher temperature in the model (Fig 6 (b) and (d)). As the yield of CINO2 and N2O5 uptake coefficient are held constant, the modelled CINO2 concentrations are not sensitive the change of RH.

14. Line 691: Please provide a proper reference here rather than citing the general website of IUPAC

Answer: Done.

15. Supporting Information Figure S2: Explain why the response of CINO2 decrease with H2O concentration (a)? Show the correlation coefficient for this linear fitting (b) as the points are spreading wide from the fitted-line.

**Answer:** We added the linear correlation coefficient ( $R^2$ ) to Figure S2. We also added a sentence in the caption to explain the decreasing trend of CINO2 signal "The decreasing trend of the CINO2 signal with increasing humidity reflects the fact that the reaction of CINO2 with higher-order clusters of  $I^{-}(H_2O)_n$  is slower than that with  $I^{-}(H_2O)$  alone (Kercher et al., 2009;Slusher et al., 2004)."

**References**

- Brown, S. S., and Stutz, J.: Nighttime radical observations and chemistry, Chem. Soc. Rev., 41, 6405-6447, 10.1039/c2cs35181a, 2012.
- Kercher, J., Riedel, T., and Thornton, J.: Chlorine activation by N2O5: simultaneous, in situ detection of CINO2 and N2O5 by chemical ionization mass spectrometry, Atmos. Meas. Tech., 2, 193-204, 10.5194/amt-2-193-2009, 2009.
- Mielke, L. H., Furgeson, A., Odame-Ankrah, C. A., and Osthoff, H. D.: Ubiquity of CINO2 in the urban boundary layer of Calgary, Alberta, Canada, Can. J. Chem., 94, 414-423, 10.1139/cjc-2015-0426, 2016.

- Slusher, D. L., Huey, L. G., Tanner, D. J., Flocke, F. M., and Roberts, J. M.: A thermal dissociation–chemical ionization mass spectrometry (TD-CIMS) technique for the simultaneous measurement of peroxyacyl nitrates and dinitrogen pentoxide, Journal of Geophysical Research: Atmospheres, 109, 10.1029/2004jd004670, 2004.
- Sommariva, R., Hollis, L. D. J., Sherwen, T., Baker, A. R., Ball, S. M., Bandy, B. J., Bell, T. G., Chowdhury, M. N., Cordell, R. L., Evans, M. J., Lee, J. D., Reed, C., Reeves, C. E., Roberts, J. M., Yang, M., and Monks, P. S.: Seasonal and geographical variability of nitryl chloride and its precursors in Northern Europe, Atmos. Sci. Lett., 19, e844, 10.1002/asl.844, 2018.